# A Compound Containing Aldehyde Dehydrogenase Relieves the Effects of Alcohol Consumption and Hangover Symptoms in Healthy Men: An Open-Labeled Comparative Study

**DOI:** 10.3390/ph17081087

**Published:** 2024-08-20

**Authors:** In-Kyung Jeong, Anna Han, Ji Eun Jun, You-Cheol Hwang, Kyu Jeung Ahn, Ho Yeon Chung, Bo Seung Kang, Se-Young Choung

**Affiliations:** 1Division of Endocrinology and Metabolism, Department of Internal Medicine, Kyung Hee University Hospital at Gangdong, Kyung Hee University School of Medicine, Seoul 05278, Republic of Koreaahnkj@khu.ac.kr (K.J.A.);; 2Department of Food Science and Human Nutrition, Jeonbuk National University, Jeonju 54896, Republic of Korea; annahan8659@jbnu.ac.kr; 3K-Food Research Center, Jeonbuk National University, Jeonju 54896, Republic of Korea; 4Department of Emergency Medicine, Hanyang University Guri Hospital, Hangyang University College of Medicine, Guri 11923, Republic of Korea; 5Department of Preventive Pharmacy and Toxicology, College of Pharmacy, Kyung Hee University, 26, Kyungheedae-ro, Dongdaemun-gu, Seoul 02453, Republic of Korea; 6Department of Pharmacy, College of Pharmacy, Dankook University, Cheonan 31116, Republic of Korea

**Keywords:** alcohol drinking, genetic polymorphism, alcohol dehydrogenase, aldehyde dehydrogenase

## Abstract

This open-labeled and comparative study aimed to test the efficacy and safety of a fermented rice extract-based substance containing yeast-fermented powder having aldehyde dehydrogenase (KisLip^®^, Pico Entech, Republic of Korea) in healthy male individuals. Healthy male subjects (n = 20) consumed 90 g of alcohol at their first visit. At the second visit, participants consumed 90 g of alcohol or alcohol with a low dose of KISLip^®^ (2000 mg, KL-L) and then 90 g of alcohol or alcohol with a high dose of KISLip^®^ (3000 mg, KL-H) at the third visit. The efficacy of KISLip^®^ depends on the mutational status of important genes related to alcohol metabolism, including alcohol dehydrogenase (*ADH1B*), cytochrome P4502E1 (*CYP2E1* (*5B*) and *CYP2E1* (*6*)), and aldehyde dehydrogenase (*ALDH2*). KISLip^®^ significantly reduced the highest level (Cmax) of alcohol and overall levels of acetaldehyde compared to the alcohol-only group in a dose-dependent manner. These significant effects of KISLip^®^ on alcohol metabolism were observed independent of mutations in the four genes. In addition, hangover symptoms were significantly decreased in the KISLip^®^ treated groups. During the study, the participants did not show any adverse events after KISLip^®^ intake. This clinical study suggested that supplementation of KISLip^®^ had beneficial effects on alcohol metabolism and might ameliorate the severity of hangovers without any adverse events.

## 1. Introduction

The most recent definition of alcohol hangovers is “the combination of negative mental and physical symptoms experienced after a single episode of alcohol consumption” [1]. The symptoms of hangovers differ depending on the individual and drinking episodes, but they commonly include tiredness, thirst, headache, nausea, and reduced physical function [2]. Following the increase in hangover incidence, alcohol consumers have shown a strong need for efficacious hangover treatments [3]. For instance, approximately 75% of alcohol drinkers consuming light-to-moderate amounts of alcohol reported the occurrence of hangovers at least once [4].

Alcohol degradation occurs in the liver primarily through two key enzymes, alcohol dehydrogenase (ADH) and aldehyde dehydrogenase (ALDH) [5]. ADH oxidizes alcohol to acetaldehyde, and ALDH oxidizes acetaldehyde to acetate and water. Additionally, chronic and/or high alcohol consumption activates the microsomal ethanol degradation pathway, which is catalyzed by cytochrome P4502E1 (CYP2E1) [6,7]. The ability of an individual to metabolize alcohol is directly associated with the incidence and severity of hangovers, as the accumulation of alcohol and acetaldehyde leads to hangovers [7,8]. Indeed, the rapid degradation of alcohol lowers the experience of alcohol hangovers and/or their severity [9,10]. Therefore, treatments that result in rapid alcohol metabolism could effectively reduce the experience and/or severity of alcohol hangovers.

The direct correlation between alcohol metabolism genetics and hangover prevalence is not yet fully understood, but genetic variations in ADH and ALDH influence alcohol metabolism [6]. For instance, individuals with *ADH1B*2*, *ADH1B*3*, and *ADH1C*1* alleles show relatively faster alcohol catabolism to acetaldehyde, whereas individuals with *ADH1B*1* and *ADH1C*2* alleles show relatively slower alcohol degradation [6]. These observations imply that the incidence, severity, and treatment outcomes of hangovers could differ based on the mutational status of the key enzymes involved in alcohol metabolism.

The anti-hangover effects of diverse foods, nutritional compounds, and/or company products have been investigated at the pre-clinical level [11,12]; few clinical trials have evaluated their safety and efficacy [13,14,15]. Red ginseng strongly ameliorates the effects of alcohol consumption and hangover symptoms in healthy men [14]. The extract of the oriental raisin tree improves alcohol hangovers; interestingly, its efficacies differ based on the presence of *CYP2E1* polymorphisms [13]. KISLip^®^ (Pico Entech, Republic of Korea) is a Saccharomyces cerevisiae fermented rice extract-based substance containing yeast-fermented powder with overexpressed ALDH [12,16]. According to the previous study, KISLip^®^ increases the metabolic capacity of ALDH2 and reduces serum acetaldehyde levels in an in vivo model, showing the potential for an efficient alcohol hangover relief treatment [12]. However, the efficacy and safety of KISLip^®^ have not yet been clinically explored. This study aimed to evaluate the efficacy and safety of KISLip^®^ supplementation in healthy male subjects. In addition, the effects of KISLip^®^ were evaluated relative to the mutational status of the genes related to alcohol metabolism.

## 2. Results

### 2.1. Study Design and Baseline Characteristics of Subjects

The study design is shown in Figure 1. A total of 29 individuals were recruited, and three subjects were excluded owing to withdrawal of consent. A total of 26 individuals were enrolled, and four individuals dropped out during the trial because of vomiting after alcohol intake (n = 3) and a highly lipemic blood sample (n = 1). The baseline characteristics of subjects were (1) all-male individuals; (2) age, 34.0 ± 5.88 years; (3) body weight, 77.2 ± 11.3 kg; and (4) body mass index (BMI), 25.3 ± 3.5 kg/m^2^.

### 2.2. Efficacy of KISLip^®^ Based on Alcohol Metabolism

Blood alcohol and acetaldehyde levels were measured at 0, 0.25, 0.5, 1, 2, 4, 6, and 15 h after alcohol consumption. The highest level (Cmax) of alcohol in all groups was observed one hour after alcohol consumption (Figure 2A). Compared to the EtOH group (7.58 ± 0.53 g/L), both KL-L (5.55 ± 0.23 g/L) and KL-H (4.18 ± 0.14 g/L) groups showed significantly decreased Cmax levels of alcohol (Figure 2B). Moreover, the KL-H group had a markedly lower Cmax than the KL-L group. Total alcohol levels at 15 h were calculated using the area under the curve (AUC). The KL-H group had significantly lower AUC (25.27 ± 2.12) compared to both the EtOH (30.85 ± 2.67) and KL-L (29.7 ± 1.91) groups, whereas the KL-L group did not show any changes in AUC compared to the EtOH group (Figure 2C).

The EtOH and KL-L groups showed the highest levels (Cmax) of acetaldehyde at 6 h, whereas the KL-H group showed Cmax at 4 h after alcohol consumption (Figure 2D). The KL-L (1.2 ± 0.15 mg/L) and KL-H (1.3 ± 0.35 mg/L) groups showed a decreasing trend in Cmax levels of acetaldehyde compared to the EtOH group (1.65 ± 0.17 mg/L) (Figure 2E). The total acetaldehyde levels for 15 h were analyzed by calculating the ACU. The KL-L (9.39 ± 1.07) and KL-H (5.22 ± 0.99) groups had significantly lower AUC compared to the EtOH group (13.02 ± 1.18), and the KL-H group had a reduced AUC compared to the KL-L group (Figure 2F).

Experiences of alcohol hangovers were evaluated through a survey conducted after 2 and 15 h of alcohol consumption. The hangover symptom scores of the participants in the trials were analyzed (Table 1). Compared with the EtOH group, fatigue and insomnia symptoms after 2 h of alcohol consumption were significantly improved in the KL-L and KL-H groups. Excessive thirst, sleepiness, fatigue, loss of concentration, increase in sensitivity, and depression symptoms after 15 h of alcohol consumption in the KL-L and KL-H groups were markedly improved compared to those in the EtOH group.

### 2.3. Effects of KISLip^®^ in the Context of Mutations in Alcohol Metabolism-Related Genes

To investigate the effects of the mutational status on alcohol metabolism-related genes, mutations in four genes involved in alcohol degradation were analyzed: alcohol dehydrogenase (*ADH1B*), two isozymes of cytochrome P4502E1, *CYP2E1* (*5B*) and *CYP2E1* (*6*), and aldehyde dehydrogenase (*ALDH2*). A total of 15 of 22 subjects had at least one gene mutation, and six subjects had more than two gene mutations. The mutation frequencies in *ADH1B*, *CYP2E1* (*6*), *CYP2E1* (*5B*), and *ALDH2* were 45.5%, 31.8%, 22.7%, and 13.6%, respectively (Figure 3A).

Each gene mutational status was dependent on Cmax levels of alcohol and acetaldehyde, and the AUC of alcohol and acetaldehyde were analyzed to scrutinize the effects of each gene mutation on alcohol metabolism. Overall, there were no statistically significant differences in the aforementioned parameters regarding mutations in *ADH1B*, *CYP2E1* (*6*), *CYP2E1* (*5B*), or *ALDH2* (Table 2, Table 3, Table 4 and Table 5). However, the KL-L group, with a *CYP2E1* (*5B*) mutation, showed significantly higher acetaldehyde Cmax levels than those without, whereas the EtOH and KL-H groups did not show any marked differences regarding *CYP2E1* (*5B*) mutations (Table 4).

Next, the Cmax levels of alcohol and acetaldehyde and AUC of alcohol and acetaldehyde were further analyzed based on the individual’s mutational status by dividing the samples into two groups: mutant group with at least one gene mutation and wild-type groups without any mutations. There were no statistically significant differences in any of the values between the mutant and wild-type groups (Table 6).

### 2.4. Efficacy of KISLip^®^ Based on Genetic Mutation

To assess the efficacy of KISLip^®^ based on mutational status, the aforementioned indicators were compared among the groups. In both the mutant and wild-type status, the KL-H group had significantly lower alcohol Cmax levels than the EtOH and KL-L groups (Figure 3B, upper panel). Furthermore, the alcohol Cmax in the KL-L group was markedly lower than that in the EtOH group with genetic mutations. Overall, the AUC of alcohol showed a KISLip dose-dependent reduction trend in both the mutant and wild-type conditions (Figure 3B, bottom panel). With respect to the mutational status, the KL-H group had a significantly reduced alcohol AUC compared to the EtOH group. There were no significant differences in the Cmax of acetaldehyde within each group (Figure 3C, upper panel), whereas the AUC of acetaldehyde tended to be lower in both the KL-L and KL-H groups than in EtOH in the mutant group but without statistical significance (Figure 3C, bottom panel). In the wild-type status, only the KL-H group showed a decreased tendency in the AUC of acetaldehyde compared to the EtOH group.

### 2.5. Assessment of Safety

The signs and symptoms related to KISLip^®^ intake were investigated by a questionnaire and investigators. None of the participants reported any adverse symptoms. Additionally, laboratory examinations were performed, and no difference in systolic or diastolic blood pressure, AST, ALT, γ-glutaryl transpeptidase (γ-GT), amylase, lipase, fasting glucose, or creatinine levels was observed before and after alcohol consumption during the study period (Table 7). The KL-L and KL-H groups showed no marked changes in AST, ALT, or γ-GT levels between baseline and 15 h after alcohol consumption (Table 8).

## 3. Discussion

Alcohol metabolism dysfunction is considered an important causal factor of alcohol hangovers [8]. An association between quicker alcohol elimination and lower severity and/or rate of hangovers has been reported [9,10]. Although there are several anti-hangover products currently available, concrete scientific evidence of their efficacy and safety is insufficient. The present study investigated the efficacy and safety of fermented rice powder-based anti-hangover product, KISLip^®^ (Pico Entech, Republic of Korea), in healthy male individuals. The current open-labeled and comparative study demonstrated that supplementation of KISLip^®^ reduced the blood levels of alcohol and acetaldehyde in healthy male subjects. KISLip^®^ intake also reduced the Cmax and AUC of alcohol and acetaldehyde levels. Additionally, hangover symptoms were significantly decreased in KISLip^®^ treated groups. There were no significant differences in the efficacy of KISLip^®^ regarding the subject’s mutational status in *ADH1B*, *CYP2E1* (*6*), *CYP2E1* (*5B*), or *ALDH2*. Lastly, no adverse events associated with KISLip^®^ were observed during the trial.

KISLip^®^ improves the metabolic capacity of ALDH2 in vivo and lowers serum acetaldehyde concentration [12], suggesting its potential for alcohol hangover mitigation. Consistent with previous observations, the present study found that intake of KISLip^®^ significantly improved blood alcohol and acetaldehyde levels in healthy human subjects. This study also observed that intake of KISLip^®^ significantly alleviates the Cmax of alcohol levels and alcohol AUC in the mutant group. Since KISLip^®^ contains ALDH activity, we hypothesized the mutational status of four pivotal genes (*ADH1B*, *CYP2E1* (*6*), *CYP2E1* (*5B*), and *ALDH2*) of the participant might influence the participant’s alcohol metabolism by KISLip^®^ intake; however, there were no differences in the alcohol metabolism of the participants based on the mutational status. Interestingly, the outcomes of *ADH1B* mutations differed based on their subvariants; for instance, compared to individuals with *ADH1B*1* subvariant mutations, people with *ADH1B*2* and *ADH1B*3* have faster alcohol metabolism and relatively lower amounts of alcohol crossing the blood–brain barrier [7]. Thus, detailed information regarding an individual’s genetic mutations is required to investigate the association between genetic mutations and hangover features, including frequency and severity.

The intake of specific nutrients and dietary components influences alcohol metabolism, affecting the experience and/or severity of hangovers [17,18]. For example, dietary intake of nicotinic acid and zinc is negatively associated with hangover severity [19], and alanine-fortified tomatoes improve alcohol hangovers [20]. These previous studies imply that differences in dietary components among participants could be a critical factor influencing hangover symptoms. Therefore, in the present study, the participants were fed a unified diet for all dinners before the start of the study. Future studies regarding the effectiveness of anti-hangover compounds need to consider the participant’s food intake history during the trial period to assess its precise efficacy. Moreover, the usual drinking amount of the individuals (e.g., g/day of alcohol) might also affect their alcohol metabolism, which needs to be investigated and considered to evaluate the effectiveness of KISLip^®^ in future studies.

The inflammatory response to alcohol plays a significant role in alcohol hangover pathology [8,21]. The severity of hangovers is markedly associated with increased levels of multiple biomarkers of the inflammatory response, such as Interleukin-6 (IL-6), IL-12, interferon-γ (IFN-γ), and tumor necrosis factor-α (TNF-α) [22,23]. Additionally, the potential functions of oxidative stress have been suggested to be involved in inflammatory responses [7,24]. For instance, oxidative stress biomarkers such as malondialdehyde and 8-isoprostrane are positively correlated with the severity of hangovers [21]. As KISLip^®^ significantly lowered alcohol and acetaldehyde levels, it will be interesting to investigate whether KISLip^®^ intake also improves the inflammatory response to alcohol consumption. According to earlier studies, a fermented rice extract-based substance containing yeast-fermented powder (the base components of KISLip^®^) shows neuroprotective effects, anti-alcohol effects, and anti-hangover effects in animal models by exhibiting antioxidant and anti-aldehyde activities due to its aldehyde-reducing compound (ARC) [12,16,25], implying that the anti-hangover outcomes of KISLip^®^ observed in this study might also come from ARC. Future clinical investigations need to study the roles of ARC in KISLip^®^’s anti-hangover outcomes.

The current study has several limitations. First, the sample size was too small to confirm our results. In particular, future studies with larger sample sizes will be necessary to confirm the results regarding the genetic association analysis of the present study. In addition, the study design was an open-labeled comparative clinical trial performed at a single center. Therefore, future randomized controlled studies are needed to test the efficacy of KISLip^®^ in larger populations.

## 4. Materials and Methods

### 4.1. Composition of KISLip^®^

KISLip^®^ (Pico Entech, Pangyo Global Biz Center, Seongnam-si, Gyeonggi-do, Republic of Korea) was produced using the fermented extract of rice. ALDH-overexpressing Saccharomyces cerevisiae was cultivated in YPD medium (2% glucose, 1% yeast extract, 2% peptone) at 30 °C and 120 rpm for the first and second rounds of fermentation. The second inoculation was performed with a fermented medium containing rice (2% glucose, 2.5% yeast extract, 0.5% peptone, 1% ammonium sulfate, 0.1% potassium diphosphate, 0.1% monopotassium phosphate, 0.5% magnesium sulfate), followed by a third round of cultivation. During this process, the cultivation temperature was maintained at 30 °C, agitation speed at 200 rpm, aeration rate at 0.5 VVM, and the pH of the culture medium was controlled at pH 5.5 using 5 N NaOH. The fermented rice culture medium, after cultivation, was filtered using a filtration system to recover solid residues. Subsequently, alcohol was added to denature the edible yeast, and the mixture was freeze-dried until the moisture content reached 10% or lower. Finally, the dried solid material was pulverized and sterilized at 60 °C, ultimately producing the final rice fermentation powder. Finally, the dried solid material was pulverized. Then, adequate ethanol was added to the pulverized power to prevent denaturation from heat shock under the sterilization conditions (60 °C), producing the final rice fermentation powder, which still has ALDH activity. The detailed composition of KISLip^®^ is shown in Appendix A.

### 4.2. Recruitment of Participants

Eligible individuals (1) were healthy male adults at the age 25~45, (2) had over 45 kg and ±20% of ideal body weight defined by (height [cm] − 100) × 0.9, (3) could drink more than 1 bottle of Soju (a Korean distilled alcohol) and had experienced hangovers, and (4) were informed and agreed to participate in the study. Females were not recruited for the current study because the menstrual cycle is associated with alcohol consumption and metabolism [26,27]. Exclusion criteria included the following: a medical history of acute disease before the start of the trial; a history of absorption, metabolism, or excretion disease (i.e., inflammatory gastrointestinal disease, gastrectomy, and liver disease); kidney, liver, endocrine or respiratory diseases; previous or current stomach and/or duodenal ulcer; aspartate transaminase (AST), alanine transaminase (ALT), and total bilirubin levels more than two-fold higher than the normal range; allergic disease; sensitivity to alcohol or alcoholic history; systolic blood pressure ≥ 140 mmHg and diastolic blood pressure ≥ 90 mmHg; eye disease such as glaucoma; participation in other clinical trials within the last 60 days; a continuous intake of caffeine (5 cups of coffee/day) or alcohol (>30 g/day), and being a smoker (10 cigarettes/day). Informed consent was obtained from all subjects.

### 4.3. Study Design

This open-labeled comparison study was performed at Kyung Hee University Hospital in Gangdong and approved by the Institutional Review Board (IRB) of Gangdong Kyung Hee University Hospital (KHNMC 2019-05-011-015). The number of the Clinical Research Information Service (CRIS) is KCT0004917 (first trial registration: 19 December 2019). All methods were performed according to the relevant guidelines and regulations. During the trial, the participants visited the hospital weekly for 3 weeks. The participants consumed the following: first week, 90 g of alcohol (450 mL of Soju) only; second week, 90 g of alcohol (450 mL of Soju) with high dose of KISLip^®^ (3000 mg = 750 mg × 4 tablets, fermented rice powder 1500 mg); and third week, 90 g of alcohol (450 mL of Soju) with low dose of KISLip^®^ (2000 mg = 2000 mg × 1 tablet, fermented rice powder 500 mg). Two hours after dinner, the subject orally ingested KISLip^®^; after 30 min of KISLip^®^ intake, the participants drank 90 g of alcohol for 30 min. After alcohol consumption, participants were allowed to consume water.

### 4.4. Efficacy Assessment

Blood samples were collected before alcohol intake and 15 min, 30 min, 1, 2, 4, 6, and 15 h after alcohol intake. Blood alcohol and acetaldehyde levels were measured at each time-point. All blood tests were performed at the Kyung Hee University Hospital in Gangdong, Seoul, Korea. The levels of alcohol (ab65343; Abcam, Cambridge, MA, USA) and acetaldehyde (#10668613035; Roche, Penzberg, Germany) were measured using an enzymatic method with a spectrophotometer in the laboratory of the Department of Pharmacology at Kyung Hee University.

Based on the previous studies [2,15], a questionnaire for hangover symptoms was established (Appendix A). Then, the survey was conducted at 2 and 15 h after administration of alcohol. The survey asked about their experience and severity of symptoms, and the scale ranged from 1 (none) to 5 (very severe). The total points were calculated, and ≥30 was defined as having hangover symptoms.

### 4.5. Assessment of Genetic Mutations

Mutations in genes related to alcohol degradation, including alcohol dehydrogenase (*ADH1B*), two isozymes of cytochrome P4502E1, *CYP2E1* (*5B*) and *CYP2E1* (*6*), and aldehyde dehydrogenase (*ALDH2*), were analyzed in blood samples at BIONEER Corporation, Daejeon, Republic of Korea.

### 4.6. Safety Assessment

The safety of KISLip^®^ intake was assessed by vital signs such as blood pressure, pulse rate, and laboratory data (hepatic function, renal function, glucose, and the pancreatic enzymes, amylase and lipase). Blood samples were collected during the trial, and the above parameters were analyzed. The participants were also asked about all adverse clinical events, including gastrointestinal and nervous system symptoms, using a questionnaire 2 and 15 h after alcohol consumption.

### 4.7. Statistical Analysis

SPSS software (version 22.0; Chicago, IL, USA) was used for the data analysis. Continuous variables were presented as means ± standard error of mean (SEM) if the results followed a normal distribution or as medians (25–75%) otherwise. To compare the differences between the two groups, either the Student’s *t*-test or Mann–Whitney U-test was used. To investigate the differences among three groups (EtOH, KL-L, and KL-H), we applied either a one-way analysis of variance (ANOVA) test when the variable exhibited a normal distribution or a Kruskal–Wallis test when the variable exhibited a skewed distribution. The statistical significance of the alcohol and acetaldehyde levels was analyzed using repeated measures analysis of variance (RM-ANOVA) or Friedman tests with the same time interval based on the sample number. If there were differences between the groups, a Bonferroni post-hoc test was performed. The symptoms of hangovers were evaluated using a questionnaire, and statistical differences were analyzed using Cochran’s Q test. *p* < 0.05 was considered significant.

## 5. Conclusions

Taken together, to our knowledge, this study is the first clinical study demonstrating the potential anti-hangover effects of KISLip^®^ in healthy individuals. Supplementation of KISLip^®^ before alcohol consumption significantly improved blood alcohol and acetaldehyde levels without any severe adverse effects. Interestingly, the usage of commercial anti-hangover remedies increases alcohol consumption among young adults due to their beliefs in the products [28], implying that inappropriate utilization of anti-hangover products might encourage alcohol consumption and its adverse health consequences. Therefore, educational warnings on anti-hangover products might be critical to improve and/or reduce alcohol abuse.

## Figures and Tables

**Figure 1 pharmaceuticals-17-01087-f001:**
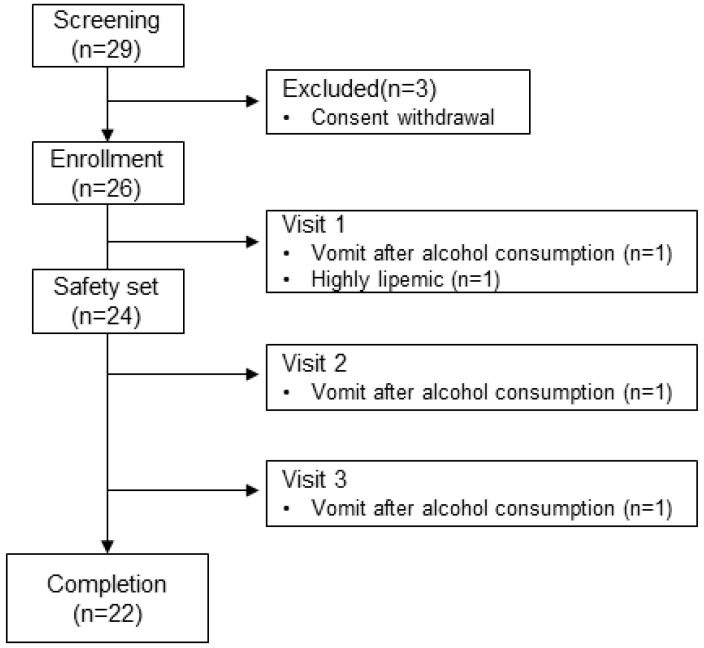
Design of the study.

**Figure 2 pharmaceuticals-17-01087-f002:**
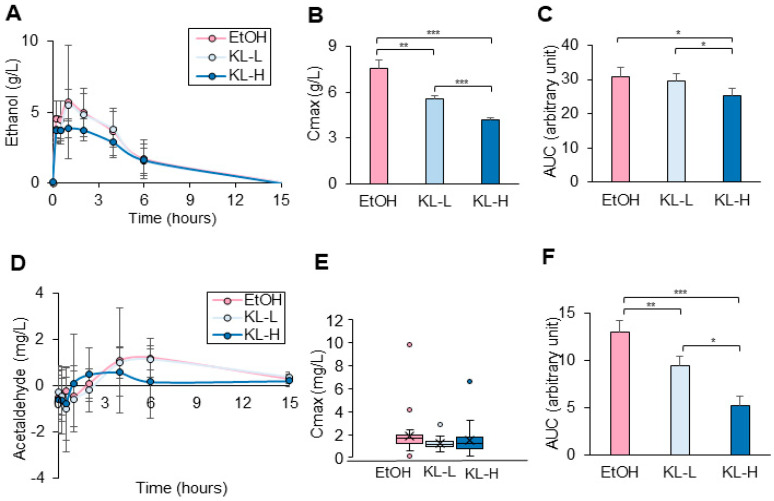
Efficacy of KISLip^®^ intake. The participants consumed 90 g alcohol (EtOH), 90 g of alcohol with low KISLip^®^ (2000 mg, KL-L), and high KISLip^®^ (3000 mg, KL-H). (**A**) Time-dependent blood alcohol level, (**B**) the highest level (Cmax) of alcohol, (**C**) overall alcohol level (AUC), (**D**) time-dependent blood acetaldehyde level, (**E**) the highest level (Cmax) of acetaldehyde and, (**F**) overall acetaldehyde level (AUC) are shown. If the results follow a normal distribution, data are shown as mean ± SEM. Otherwise, data are shown as median (25–75%). * *p* < 0.05, ** *p* < 0.01, and *** *p* < 0.001.

**Figure 3 pharmaceuticals-17-01087-f003:**
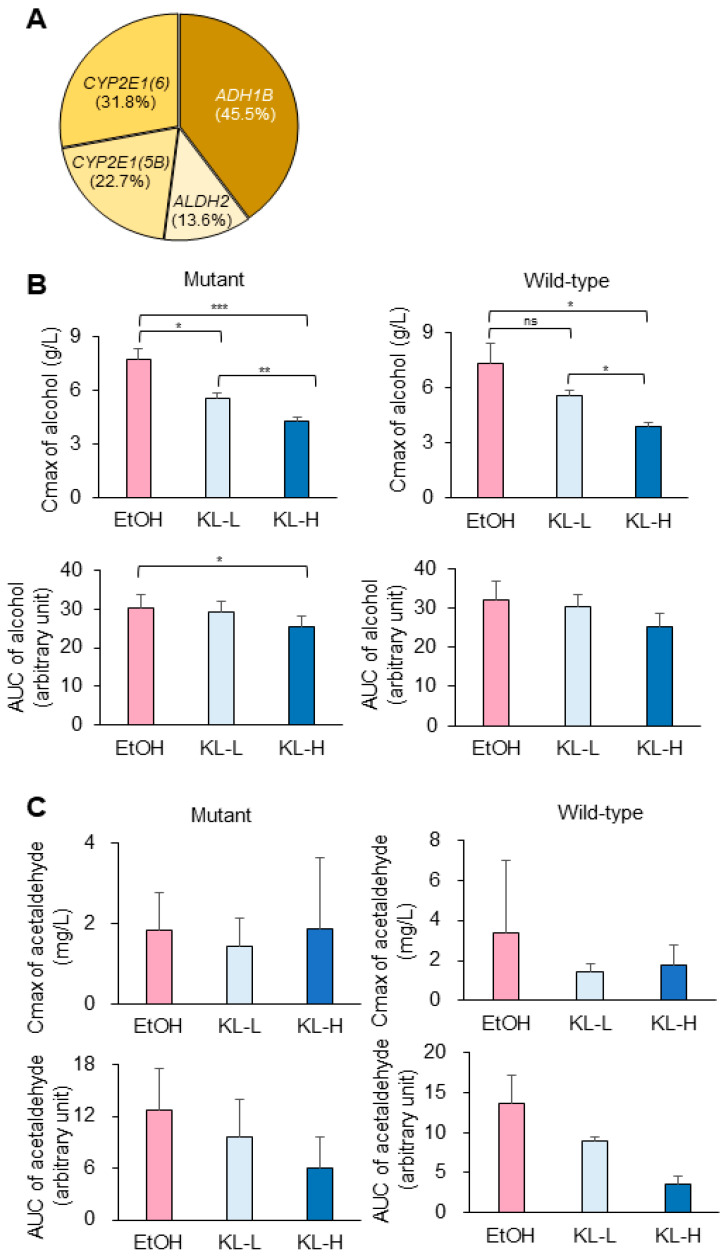
Changes in alcohol metabolism regarding mutational status. (**A**) Percentile of mutations in genes related to alcohol degradation. (**B**) The highest level (Cmax) and AUC of alcohol in mutant and wild-type groups are shown. (**C**) The highest level (Cmax) and AUC of acetaldehyde in mutant and wild-type groups are shown. The mutant group has at least one gene mutation of *ADH1B*, *CYP2E1* (*6*), *CYP2E1* (*5B*), or *ALDH2*. If the results follow a normal distribution, data are shown as mean ± SEM. Otherwise, data are shown as median (25–75%). ns, not significant, * *p* < 0.05, ** *p* < 0.01, and *** *p* < 0.001.

**Table 1 pharmaceuticals-17-01087-t001:** Hangover symptom score of participants.

Symptom	After 2 h				After 15 h			
	EtOH	KL-L	KL-H	*p*-Value	EtOH	KL-L	KL-H	*p*-Value
Excessive thirst	3.46 ± 1.10	2.82 ± 1.33	2.85 ± 1.08	0.129	3.41 ± 1.26	2.73 ± 1.24	2.46 ± 1.06	0.002
Sleepiness	3.27 ± 0.94	2.73 ± 1.12	2.64 ± 1.22	0.098	2.68 ± 1.21	2.41 ± 1.01	2.09 ± 1.11	0.016
Headache	1.86 ± 0.94	1.68 ± 0.96	1.73 ± 0.99	0.794	1.96 ± 1.09	1.77 ± 1.11	1.82 ± 1.05	0.641
Dizziness	1.96 ± 1.09	1.77 ± 1.07	1.73 ± 1.03	0.584	1.59 ± 0.85	1.59 ± 1.14	1.73 ± 1.08	0.497
Vomiting	1.46 ± 0.74	1.27 ± 0.63	1.50 ± 0.91	0.156	1.36 ± 0.90	1.41 ± 0.85	1.36 ± 0.95	0.646
Fatigue	2.14 ± 1.08	1.64 ± 0.73	1.59 ± 0.80	0.004	2.41 ± 1.22	1.77 ± 0.92	1.50 ± 0.80	0.000
Stomachache	1.36 ± 0.66	1.36 ± 0.66	1.64 ± 1.14	0.142	1.54 ± 0.96	1.50 ± 1.01	1.27 ± 0.55	0.228
Nausea	1.27 ± 0.55	1.18 ± 0.40	1.27 ± 0.63	0.368	1.32 ± 0.65	1.23 ± 0.53	1.27 ± 0.55	0.867
Loss of concentration	2.18 ± 1.18	1.82 ± 0.96	1.82 ± 1.01	0.201	2.00 ± 1.07	1.68 ± 0.89	1.46 ± 0.67	0.011
Increased sensitivity (i.e., brightness and noise)	1.86 ± 1.25	1.54 ± 0.96	1.59 ± 0.73	0.168	1.86 ± 1.21	1.27 ± 0.70	1.36 ± 0.73	0.005
Insomnia	0.96 ± 1.25	0.64 ± 0.95	1.35 ± 0.72	0.045	1.96 ± 1.21	1.36 ± 0.58	1.64 ± 1.05	0.066
Sweat	1.36 ± 0.58	1.27 ± 0.63	1.32 ± 0.65	0.756	1.36 ± 0.73	1.14 ± 0.47	1.04 ± 0.21	0.156
Depression	1.36 ± 0.66	1.32 ± 0.78	1.36 ± 0.85	0.607	1.50 ± 1.01	1.18 ± 0.50	1.09 ± 0.29	0.038
Memory loss	1.50 ± 0.91	1.09 ± 0.29	1.27 ± 0.77	0.129	1.23 ± 0.53	1.04 ± 0.21	1.04 ± 0.21	0.165

Data are shown as mean ± SEM.

**Table 2 pharmaceuticals-17-01087-t002:** *ADH1B*-dependent changes in alcohol metabolism indicators.

	Mutation (+) (n = 10)	Mutation (−) (n = 12)	*p*-Value
EtOH			
Alcohol Cmax (g/L)	6.82 ± 2.09	8.22 ± 2.70	0.195
Alcohol AUC	27.95 ± 13.17	33.27 ± 11.94	0.333
Acetaldehyde Cmax * (g/L)	1.56 (0.75–1.69)	1.71 (1.36–1.99)	0.283
Acetaldehyde AUC *	12.49 (8.40–15.43)	14.51 (9.48–16.28)	0.596
KL-L			
Alcohol Cmax (g/L)	5.60 ± 1.36	5.50 ± 0.84	0.833
Alcohol AUC	27.97 ± 10.20	31.13 ± 7.96	0.423
Acetaldehyde Cmax * (g/L)	1.07 (0.86–1.45)	1.23 (1.09–1.77)	0.228
Acetaldehyde AUC *	7.65 (5.03–12.16)	8.13 (6.72–12.23)	0.862
KL-H			
Alcohol Cmax (g/L)	4.22 ± 0.72	4.15 ± 0.64	0.793
Alcohol AUC	22.92 ± 10.21	27.23 ± 9.67	0.323
Acetaldehyde Cmax * (g/L)	1.40 (0.93–2.71)	1.18 (0.51–2.12)	0.346
Acetaldehyde AUC *	4.46 (3.67–8.99)	3.04 (0.69–7.79)	0.136

If the results follow a normal distribution, data are shown as mean ± SEM. Otherwise, the data are shown as median (25–75%). Cmax, the highest level; AUC, area under the curve. * Except for error samples with negative values (n = 15).

**Table 3 pharmaceuticals-17-01087-t003:** *ALDH2*-dependent changes in alcohol metabolism indicators.

	Mutation (+) (n = 3)	Mutation (−) (n = 19)	*p*-Value
EtOH			
Alcohol Cmax (g/L)	7.03 (2.72–7.03)	7.56 (6.10–9.58)	0.356
Alcohol AUC	42.55 (11.08–42.55)	28.13 (25.30–36.48)	0.523
Acetaldehyde Cmax * (g/L)	1.35 (0.79–1.35)	1.65 (1.28–1.97)	0.408
Acetaldehyde AUC *	10.56 (5.79–10.56)	13.47 (9.49–15.43)	0.654
KL-L			
Alcohol Cmax (g/L)	5.48 (3.76–5.48)	5.46 (4.83–6.61)	0.651
Alcohol AUC	36.13 (12.08–36.13)	29.86 (21.61–37.30)	0.929
Acetaldehyde Cmax * (g/L)	1.40 (1.07–1.40)	1.15 (0.95–1.59)	0.308
Acetaldehyde AUC *	8.78 (5.27–8.78)	7.57 (6.24–11.96)	0.887
KL-H			
Alcohol Cmax (g/L)	5.11 (4.12–5.11)	4.10 (3.57–4.51)	0.087
Alcohol AUC	40.87 (18.70–40.87)	24.94 (14.89–30.57)	0.132
Acetaldehyde Cmax * (g/L)	2.18 (1.56–2.18)	1.22 (0.85–1.66)	0.651
Acetaldehyde AUC *	8.43 (8.11–8.43)	3.92 (1.16–8.08)	0.261

If the results follow a normal distribution, data are shown as mean ± SEM. Otherwise, the data are shown as median (25–75%). Cmax, the highest level; AUC, area under the curve. * Except for error samples with negative values (n = 15).

**Table 4 pharmaceuticals-17-01087-t004:** *CYP2E1* (*5B*)-dependent changes in alcohol metabolism indicators.

	Mutation (+) (n = 5)	Mutation (−) (n = 17)	*p*-Value
EtOH			
Alcohol Cmax (g/L)	7.23 (6.37–11.39)	7.03 (5.27–9.20)	0.493
Alcohol AUC	28.13 (20.33–44.64)	29.32 (22.80–42.43)	0.820
Acetaldehyde Cmax * (g/L)	1.65 (1.10–2.11)	1.59 (1.10–1.95)	0.880
Acetaldehyde AUC *	13.47 (8.19–18.41)	12.40 (9.29–15.45)	0.924
KL-L			
Alcohol Cmax (g/L)	5.30 (4.68–5.78)	5.48 (4.77–6.63)	0.704
Alcohol AUC	29.86 (19.52–37.19)	31.57 (22.65–37.86)	0.762
Acetaldehyde Cmax * (g/L)	1.59 (1.32–2.04)	1.13 (0.91–1.30)	0.031
Acetaldehyde AUC *	10.62 (6.46–14.67)	7.41 (5.51–12.28)	0.398
KL-H			
Alcohol Cmax (g/L)	4.43 (3.79–5.51)	4.12 (3.57–4.51)	0.401
Alcohol AUC	25.02 (12.26–34.85)	24.94 (16.60–33.79)	0.762
Acetaldehyde Cmax * (g/L)	1.48 (0.54–4.10)	1.22 (0.86–2.26)	0.880
Acetaldehyde AUC *	6.35 (3.36–8.65)	3.92 (0.84–8.39)	0.505

If the results follow a normal distribution, data are shown as mean ± SEM. Otherwise, the data are shown as median (25–75%). Cmax, the highest level; AUC, area under the curve. * Except for error samples with negative values (n = 15).

**Table 5 pharmaceuticals-17-01087-t005:** *CYP2E1* (*6*)-dependent changes in alcohol metabolism indicators.

	Mutation (+) (n = 7)	Mutation (−) (n = 15)	*p*-Value
EtOH			
Alcohol Cmax (g/L)	7.23 (6.10–9.58)	7.03 (5.29–9.48)	0.837
Alcohol AUC	28.02 (20.30–50.25)	30.99 (25.30–42.37)	1.000
Acetaldehyde Cmax * (g/L)	1.41 (0.79–1.75)	1.65 (1.35–1.99)	0.237
Acetaldehyde AUC *	11.96 (7.82–15.64)	13.46 (9.76–16.36)	0.494
KL-L			
Alcohol Cmax (g/L)	5.27 (4.30–6.05)	2.48 (4.83–6.65)	0.535
Alcohol AUC	29.86 (17.44–37.59)	31.57 (26.86–37.30)	0.731
Acetaldehyde Cmax * (g/L)	1.25 (0.57–1.59)	1.15 (1.01–1.81)	0.837
Acetaldehyde AUC *	10.62 (5.27–12.36)	7.41 (6.24–12.07)	0.689
KL-H			
Alcohol Cmax (g/L)	4.43 (3.65–4.84)	4.12 (3.50–4.51)	0.447
Alcohol AUC	17.36 (14.42–39.45)	27.03 (18.70–32.51)	0.490
Acetaldehyde Cmax * (g/L)	1.58 (0.56–4.29)	1.10 (0.85–1.66)	0.237
Acetaldehyde AUC *	3.68 (0.77–7.59)	4.19 (2.39–9.02)	0.437

If the results follow a normal distribution, data are shown as mean ± SEM. Otherwise, the data are shown as median (25–75%). Cmax, the highest level; AUC, area under the curve. * Except for error samples with negative values (n = 15).

**Table 6 pharmaceuticals-17-01087-t006:** Mutation-dependent changes in alcohol metabolism indicators.

	Mutant (n = 15)	Wild-Type (n = 7)	*p*-Value
EtOH			
Alcohol Cmax (g/L)	7.69 ± 0.62	7.36 ± 1.08	0.780
Alcohol AUC	30.33 ± 3.32	31.97 ± 4.76	0.783
Acetaldehyde Cmax * (g/L)	1.48 (1.93–1.75)	1.93 (1.58–1.99)	0.310
Acetaldehyde AUC *	12.75 ± 1.50	14.54 (9.29–17.38)	0.859
KL-L			
Alcohol Cmax (g/L)	5.54 ± 0.30	5.57 ± 0.33	0.956
Alcohol AUC	29.32 ± 2.55	30.50 ± 2.74	0.781
Acetaldehyde Cmax * (g/L)	1.20 (1.05–1.81)	1.26 (1.15–1.81)	0.679
Acetaldehyde AUC *	9.59 ± 1.36	7.34 (6.83–11.96)	0.768
KL-H			
Alcohol Cmax (g/L)	4.30 ± 0.17	3.92 ± 0.23	0.215
Alcohol AUC	25.26 ± 2.76	25.29 ± 3.32	0.996
Acetaldehyde Cmax * (g/L)	1.30 (0.95–2.34)	1.10 (0.51–2.37)	0.768
Acetaldehyde AUC *	6.01 ± 0.11	2.04 (0.69–7.41)	0.254

The mutant group contained at least one mutation in *ADH1B*, *CYP2E1* (*6*), *CYP2E1* (*5B*), or *ALDH2*. If the results follow a normal distribution, data are shown as mean ± SEM. Otherwise, the data are shown as median (25–75%). Cmax, the highest level; AUC, area under the curve. * Except for error samples with negative values (n = 15).

**Table 7 pharmaceuticals-17-01087-t007:** Laboratory results of safety assessment before and after alcohol consumption.

	Before	After	*p*-Value *
Systolic blood pressure (mmHg)	128.0 ± 2.1	125.9 ± 1.6	0.407
Diastolic blood pressure (mmHg)	78.6 ± 1.7	76.6 ± 1.5	0.232
AST (U/L)	25.7 ± 1.4	25.9 ± 2.5	0.786
ALT (U/L)	24.7 ± 2.2	22.8 ± 2.2	0.317
γ-GT (U/L)	33.0 ± 4.4	30.2 ± 4.2	0.134
Amylase (U/L)	55.0 ± 3.6	58.8 ± 3.8	0.158
Lipase (U/L)	19.0 ± 1.8	21.0 ± 2.6	0.257
Fasting glucose (mg/dL)	101.1 ± 2.2	99.5 ± 1.6	0.332
Creatinine (mg/dL)	0.84 ± 0.02	0.87 ± 0.03	0.076

AST, aspartate transaminase; ALT, alanine transaminase; and γ-GT, γ-glutaryl transpeptidase. Before, before the trial; after, after the trial. * paired comparison. Data are shown as mean ± SEM.

**Table 8 pharmaceuticals-17-01087-t008:** Hepatic function in the safety assessment before and after alcohol consumption in each group.

		EtOH (1st Trial)	EtOH (2nd Trial)	KL-L	KL-H
AST (U/L)	0 h	27.1 ± 1.6	25.9 ± 2.5	25.3 ± 1.2	22.3 ± 0.9
	15 h	23.3 ± 1.1	25.0 ± 2.3	24.8 ± 1.2	23.4 ± 1.2
	*p*-value	0.001	0.547	0.420	0.174
ALT (U/L)	0 h	28.0 ± 3.0	22.8 ± 2.2	26.1 ± 2.6	22.3 ± 1.9
	15 h	25.7 ± 2.9	22.7 ± 2.8	26.6 ± 2.6	24.1 ± 2.2
	*p*-value	0.001	0.932	0.316	0.013
γ-GT (U/L)	0 h	34.1 ± 4.0	30.2 ± 4.2	33.7 ± 4.7	32.0 ± 4.4
	15 h	32.6 ± 3.8	30.6 ± 4.4	32.9 ± 4.6	31.3 ± 4.6
	*p*-value	0.001	0.261	0.070	0.126

AST, aspartate transaminase; ALT, alanine transaminase; and γ-GT, γ-glutaryl transpeptidase. Data are shown as mean ± SEM.

## Data Availability

Data described in the manuscript, code book, and analytic code will not be made available because the Institutional Review Board (IRB) of Gangdong Kyung Hee University Hospital did not allow sharing.

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
