# Peer review of "A Compound Containing Aldehyde Dehydrogenase Relieves the Effects of Alcohol Consumption and Hangover Symptoms in Healthy Men: An Open-Labeled Comparative Study"

_pharmaceuticals, 2024, doi:10.3390/ph17081087_

Round 1

Reviewer 1 Report

Comments and Suggestions for Authors

The paper by  Jeong et al. is devoted to studying the effects of yeast-fermented powder having aldehyde dehydrogenase (KisLip®, Pico Entech, Republic of Korea) alcohol metabolism and hangover symptoms in healthy men. The paper is  structured, the figures and tables complement the text, the reference list covers the relevant literature adequately.   

However, I have some of remarks:

1. I don't see much point in doing genetic association analysis with such sample sizes. For this kind of research, the group size should be an order of magnitude larger. Judging by the conclusion, the authors understand this.

2. Can the authors speculate in the Discussion section which components of theKisLip® may be responsible for the effects of this drug on behavior and alcohol metabolism?

Author Response

    1. I don't see much point in doing genetic association analysis with such sample sizes. For this kind of research, the group size should be an order of magnitude larger. Judging by the conclusion, the authors understand this.

    Response: Thank you for the insightful comments. Now, we have addressed your comments in the Discussion section as one of the limitations of our study (Lines 289-290)

    1. Can the authors speculate in the Discussion section which components of the KisLip® may be responsible for the effects of this drug on behavior and alcohol metabolism?

    Response: We now have addressed this point in the Discussion section (Lines 72-77, 281-287).

Reviewer 2 Report

Comments and Suggestions for Authors

.The manuscript, with reference number pharmaceuticals-3139096, titled “Aldehyde dehydrogenase-related compounds relieve the effects of alcohol consumption and hangover symptoms in healthy men: An open-labeled comparative study” by In-Kyung Jeong et al., is a study on the oxidative metabolism of alcohol in a group of healthy male humans and how this metabolism may be affected by ingesting a complex natural mixture extract named KISLip®. It also considers the possible role of genetic variations on critical genes related to oxidative alcohol metabolism.

The work appears sound and relevant.

To make this manuscript suitable for publication, I would suggest considering and rectifying or explaining the following:

-The title asserts that ALDH-related compounds relieve… To what compounds do the authors refer? Why are these compounds ALDH-related? Please clarify.

-KISLip® is a mixture of compounds. Would it be possible to ascertain or speculate which specific compound or combination is responsible for the observed effects? Does it contain ALDH activity per se, or can some of the components accelerate the individuals' oxidative metabolism?

-Apparently, ALDH is present in the extract. Since the extract has been treated and sterilized at 60ºC, is there any ALDH remanent activity in the extract?

-Concerning the cohort of participants, the study establishes that the subjects could drink a bottle of Soju and have experienced hangovers in the past. This means that they are alcohol drinkers; therefore, it would be appreciated if the drinking habits were more detailed (for example, if available, how many grams of alcohol they consumed per day) since this fact may affect their metabolism outcomes. A comment on this subject would be appreciated if detailed information is unavailable.

-What comparison do the p values in the table refer to? (since there are three experimental groups).

-Tables 2, 3, and 4 should indicate units for Cmax values for ethanol and acetaldehyde.

-Was the questionnaire validated (applied to more extensive samples) to assess hangovers? For example, how was the cut-off point above 30 points determined (criteria)?

-It would be appreciated if the authors could briefly justify experimental groups' treatment timeline and choices. Why did the treatments start 30 minutes after dinner? Why is the amount of alcohol and KISLip® (in two dosages) given equally to all subjects without considering their body mass? Why were all subjects in this study male? A comment in this line would be interesting in this manuscript

-What specific mutations were found in the genes analyzed in the individuals studied? Do they correspond to previously described variants?

-What possible effects of KISLip® on alcohol metabolism in genetic variants studied would the authors expect? (Please briefly refer to it in the discussion section.)

Exploring the mechanisms and effects of ethanol metabolism is very interesting; this study is valuable in this context. However, the possible extension of this information to the general public may encourage using a commercial product to combat hangovers and drinking more alcohol without worries concerning hangover symptoms (ignoring other damages for which alcohol ingestion is responsible). Hangovers and other consequences of alcohol drinking can be avoided if drinking ceases. A brief statement concerning this matter in their manuscript would seem valuable if the authors agree.

Author Response

  1. The title asserts that ALDH-related compounds relieve… To what compounds do the authors refer? Why are these compounds ALDH-related? Please clarify.

Response: Thank you for the critical points of view. Based on your comment, we now clarified and revised our research title. Moreover, the detailed information on the KISLip® is now addressed in Lines 72-77.

  1. KISLip® is a mixture of compounds. Would it be possible to ascertain or speculate which specific compound or combination is responsible for the observed effects? Does it contain ALDH activity per se, or can some of the components accelerate the individuals' oxidative metabolism?

Response: We now have addressed this point in the Discussion section (Lines 72-77, 281-287).

  1. Apparently, ALDH is present in the extract. Since the extract has been treated and sterilized at 60ºC, is there any ALDH remanent activity in the extract?

Response: The fermented rice was pulverized. Then adequate ethanol was added to the pulverized power to make the soup to prevent denaturation from heat shock under the sterilization at 60°C. Therefore, the extract still has ALDH activity. This detailed information is now described in the Materials and Methods section (Lines 309-312).

  1. Concerning the cohort of participants, the study establishes that the subjects could drink a bottle of Soju and have experienced hangovers in the past. This means that they are alcohol drinkers; therefore, it would be appreciated if the drinking habits were more detailed (for example, if available, how many grams of alcohol they consumed per day) since this fact may affect their metabolism outcomes. A comment on this subject would be appreciated if detailed information is unavailable.

Response: Thank you for the valuable points. As detailed information is unavailable, this point is now addressed in the Discussion section (Lines 268-271).

  1. What comparison do the p values in the table refer to? (since there are three experimental groups).

Response: Thank you for pointing this out. The detailed information regarding p-values in the table has now been addressed in Lines 375-378.

  1. Tables 2, 3, and 4 should indicate units for Cmax values for ethanol and acetaldehyde.

Response: Now, all tables include units for Cmax values.

  1. Was the questionnaire validated (applied to more extensive samples) to assess hangovers? For example, how was the cut-off point above 30 points determined (criteria)?

Response: As we know, there is no validated questionnaire for hangover assessment yet. We now cited references that were used to establish the questionnaire of the present study and addressed this point in the Material and Method section > 4.4 Efficacy assessment (Lines 353-354).   

  1. It would be appreciated if the authors could briefly justify experimental groups' treatment timelines and choices. Why did the treatments start 30 minutes after dinner? Why is the amount of alcohol and KISLip® (in two dosages) given equally to all subjects without considering their body mass? Why were all subjects in this study male? A comment in this line would be interesting in this manuscript

Response: Thank you for the critical point of view. First of all, since alcohol metabolism can be influenced by food intake and digestion, we started treatment 30 minutes after dinner. Moreover, as KISLip® is a functional food product (not a drug), it was not necessary to take based on BMI. Indeed, the amounts of anti-hypertension and anti-diabetic drugs are not based on the patient’s BMI, especially with adult patients. Lastly, we did not recruit female participants because the menstrual cycle affects alcohol metabolism and consumption [1,2], which is now addressed in the Material and Methods section > 4.2 Recruitment of participants (Lines 318-320).

  1. What specific mutations were found in the genes analyzed in the individuals studied? Do they correspond to previously described variants?

Response: According to the previous studies regarding the genetic polymorphism of ALDH2 and CYP2E1 in Korean adults, the allele frequencies of ALDH2(1) and ALDH2(2) were 0.840 and 0.160, respectively [3]. Therefore, the present study includes these two genes and ADH to investigate the association between the mutational status and the effects of KISLip®.  

  1. What possible effects of KISLip® on alcohol metabolism in genetic variants studied would the authors expect? (Please briefly refer to it in the discussion section.)

Response: We addressed your point in the Discussion section (Lines 249-253).

  1. Exploring the mechanisms and effects of ethanol metabolism is very interesting; this study is valuable in this context. However, the possible extension of this information to the general public may encourage using a commercial product to combat hangovers and drinking more alcohol without worries concerning hangover symptoms (ignoring other damages for which alcohol ingestion is responsible). Hangovers and other consequences of alcohol drinking can be avoided if drinking ceases. A brief statement concerning this matter in their manuscript would seem valuable if the authors agree.

Response: Thank you for your interesting point of view. Indeed, we found that the usage of hangover remedies is positively associated with alcohol consumption patterns among young adults [4]. We now addressed your point and this article in the Conclusion section (Lines 338-393).

References

  1. Carroll, H.A., M.K. Lustyk, and M.E. Larimer, The relationship between alcohol consumption and menstrual cycle: a review of the literature. Arch Womens Ment Health, 2015. 18(6): p. 773-81.
  2. Erol, A., et al., Sex hormones in alcohol consumption: a systematic review of evidence. Addiction biology, 2019. 24(2): p. 157-169.
  3. Lee, K.H., et al., Genetic polymorphism of cytochrome P-4502E1 and mitochondrial aldehyde dehydrogenase in a Korean population. Alcohol Clin Exp Res, 1997. 21(6): p. 953-6.
  4. Han, D.H., et al., Association of over the counter "hangover remedy" use with alcohol use problems and consumption pat-terns among young adults. Alcohol Alcohol, 2024. 59(2).
